# Differences and Similarities in Epidemiology and Risk Factors for Cutaneous and Uveal Melanoma

**DOI:** 10.3390/medicina59050943

**Published:** 2023-05-14

**Authors:** Daciana Elena Brănișteanu, Elena Porumb-Andrese, Alexandra Stărică, Anca Catalina Munteanu, Mihaela Paula Toader, Mihail Zemba, Vlad Porumb, Mihai Cozmin, Andreea Dana Moraru, Alin Codruț Nicolescu, Daniel Constantin Brănișteanu

**Affiliations:** 1Dermatology Department, ‘Grigore T. Popa’ University of Medicine and Pharmacy, 700115 Iasi, Romania; 2Railway Clinical Hospital, 700506 Iasi, Romania; 3Ophthalmology Department, ‘Carol Davila’ University of Medicine and Pharmacy, 050474 Bucharest, Romania; 4Department Surgery, ‘Grigore T. Popa’ University of Medicine and Pharmacy, 700111 Iasi, Romania; 5Clinical Department, Apollonia University of Iasi, 700511 Iasi, Romania; 6Ophthalmology Department, ‘Grigore T. Popa’ University of Medicine and Pharmacy, 700115 Iasi, Romania; 7Roma Medical Center for Diagnosis and Treatment, 011773 Bucharest, Romania; 8“Agrippa Ionescu” Emergency Clinical Hospital, 011773 Bucharest, Romania

**Keywords:** uveal melanoma, cutaneous melanoma, atypical mole syndrome

## Abstract

Both cutaneous melanoma (CM) and uveal melanoma (UM) represent important causes of morbidity and mortality. In this review, we evaluate the available knowledge on the differences and similarities between cutaneous melanoma and uveal melanoma, focusing on the epidemiological aspects and risk factors. Uveal melanoma is a rare condition but is the most prevalent primary intra-ocular malignant tumor in adults. Cutaneous melanoma, on the other hand, is significantly more common. While the frequency of cutaneous melanoma has increased in the last decades worldwide, the incidence of uveal melanoma has remained stable. Although both tumors arise from melanocytes, they are very distinct entities biologically, with complex and varied etiologies. Both conditions are encountered more frequently by individuals with a fair phenotype. ultraviolet-radiation is an important, well-documented risk factor for the development of CM, but has shown not to be of specific risk in UM. Although cutaneous and ocular melanomas seem to be inherited independently, there are reported cases of concomitant primary tumors in the same patient.

## 1. Introduction

Melanoma is a malignant tumor that originates from melanocytes. Cutaneous melanoma (CM), the most common type of melanoma, develops in the outermost layer of the skin, while uveal melanoma (UM), a less common type, occurs in the uvea layer of the eye [1]. Despite their anatomical differences, both types of melanomas share similar risk factors, including fair skin phenotypes and genetic predisposition. However, there are distinct epidemiological differences between the two subtypes, including age of onset, gender distribution, and survival rates. In this article, we will examine the similarities and differences between cutaneous and uveal melanoma with regards to their epidemiology and risk factors. Understanding their specificities can help identify individuals who may be at a higher risk for developing the disease, implement potential preventive measures to decrease the incidence of the disease, and enable researchers to develop more effective treatments tailored to each subtype’s unique characteristics, ultimately improving patient outcomes.

## 2. Methods

For this narrative review, the authors conducted an extensive literature search in the Medline electronic database, using the PubMed interface. The keyword combinations used were ‘uveal melanoma’, ‘cutaneous melanoma’, and, in turn, each of the following: ‘epidemiology’; ‘host susceptibility factors’; ‘environmental risk factors’. After filters were applied (case report, classical article, guide-line, journal article, meta-analysis, observational study, review, systematic review), a consistent number of references resulted. In addition to the search, we conducted a manual review of the reference lists of the articles that were included. Among these, 90 references were cited in this review.

## 3. Results

### 3.1. The Cellular Function of Melanocytes

Melanocytes are cells that produce melanin pigments in organelles called melanosomes via an enzymatic cascade that includes tyrosinase, tyrosinase-related protein-1 (TYRP1), and TYRP2/DCT (dopachrome tautomerase). The brown/black pigment (eumelanin) is generated, as well as an orange/yellow pigment (pheomelanin), which is synthesized in the presence of cysteine or glutathione. The ratio of these two forms of melanin determines skin and eye color polymorphism. Dark brown skin and eyes have a substantially higher eumelanin/pheomelanin ratio than pale skin and eyes with light-colored irises (blue, green, yellow-brown, and hazel eyes) [2,3].

Melanocytes are found in many regions of the human body, including the skin, eyes, cochlea, mucosal epithelia, meninges, and heart and are derived from neural crest cells [4,5,6].

They have a well-established role and function in the skin, but not in other anatomical regions [1].

Within organelles called melanosomes, the melanocytes are responsible for the synthesis of melanin pigments. Melanocytes in the epidermis distribute these melanin-containing melanosomes to the surrounding keratinocytes. This ensures uniform pigmentation, determines the color of the skin, and protects against ultraviolet radiation (UVR)’s detrimental effects [7].

In the eye, melanocytes can be found in the conjunctiva and all parts of the uvea (iris, ciliary body, and choroid) [6]. While the role of melanocytes in the conjunctiva remains uncertain, the quantity and quality of melanin pigment in the iris determine eye color [1]. However, in contrast to the skin, the iris color is completely unaffected by sun exposure [6]. The presence of melanin in uveal melanocytes has been linked to eye protection against a variety of ocular disorders that might result in blindness, such as age-related macular degeneration and uveal melanoma [8,9]. The mechanism by which melanin mediates this protection remains unclear [1].

### 3.2. Epidemiology

The skin is the most common site of melanoma development (90% of primary melanoma cases), although this tumor can occur in any tissue containing melanocytes [10,11].

The uvea is the second most common site for melanoma and accounts for 5% of all primary melanoma cases, with UM representing the most frequently diagnosed primary intraocular malignant tumor in adults [11,12].

UM has an annual incidence of six people per million, whereas CM has an annual incidence of 12.2 to 48.1 people per 100,000 persons [13,14]. While the worldwide incidence of CM has been steadily increasing in recent decades across all continents at a more rapid rate compared to any other type of cancer, the incidence of UM has remained stable [15].

There are constant variations in the occurrence of certain diseases in different regions of the world. Regarding UM, in the US, the incidence varies from 5.1 to 6 cases per million of the population per year, being highest in the southern latitudes [16,17]. In Europe, the incidence is much higher and varies between 1.3 to 8.6 cases per million of the population per year, with eight cases per million of the population per year in Caucasians of northern European descent (Norway and Denmark) and lowest in Italy (3.3 cases per million population per year) and Spain (1.9 cases per million population per year) [18].

Similar to UM, the incidence of CM demonstrates a characteristic geographical variability [19] as well. In the US, the standardized incidence ratio for CM is 45 new cases per 100,000 inhabitants [20]. In Europe, there is a very pronounced difference in the incidence of CM among various countries, with northern countries having the highest incidence [21]. Norway has the highest incidence rate (29.6 age-standardized rates, world population (ASWR)) and Romania has the lowest (3.4 ASWR), within the European average of 13.5 (ASWR) [22].

In both CM and UM, the incidence in European countries follows a sharp north-to-south and west-to-east decreasing curve. This gradient is directly related to the protective factor of the eye pigment present in the southern population, also respecting the high exposure to ultraviolet (UV) light in lower latitudes [17,23].

Melanoma is rare in non-Whites and several studies have demonstrated that Hispanics and Asians have a lower incidence of both CM and UM while Black individuals have the lowest incidence [17,24,25].

Analyzing skin melanoma incidence data by sex, the studies noticed that women are more prevalent in young age groups, whereas men are more prevalent from the age of 55 onwards [26]. Although there is no consistent sex-related difference in UM, the age-adjusted incidence in epidemiological studies has revealed that men had an increased prevalence (5.8 per million in males compared to 4.4 per million in females) [16].

UM develops more often in older people, with an incidence rate that increases in direct proportion to age, reaching a peak incidence at the age of 70 years and reaching a plateau phase after the age of 75 [17]. While the mean age for diagnosis in Caucasians has increased from 55 to 62 years of age, UM appears to occur at a younger age in Asian countries [27], with a mean age for diagnosis varying from 45 years of age (in the Chinese population) to 55 years of age (in the Japanese population). In children, UM is unusual [11]. Furthermore, unlike UM, CM primarily affects young and middle-aged people (median age at diagnosis, 57 years). After the age of 25, the incidence increases linearly until the age of 50 and follows a decreasing curve after [28].

### 3.3. Host Susceptibility Factors

The etiology of melanoma is a complex and heterogeneous issue that involves various factors, such as environmental, phenotypic, and genetic susceptibility [1]. The major risk factors for CM include a personal and familial history of CM, many congenital or acquired melanocytic and dysplastic nevi, sun exposure, and skin reactions to sun exposure, according to the phototype’s central role in the development of melanoma [1,29,30,31].

Certain phenotypic features are largely recognized as predisposing factors for CM, regardless of family history. Such particularities include blonde or red hair, light complexion, and light eye color (especially when associated with Central or Northern European ancestry), with a consequently heightened sensitivity and an altered response to ultraviolet radiation (UVR) exposure [32,33,34]. These qualities are often associated with the inability to tan and the tendency to develop freckles or moles, and emphasize the importance of UVR in the development of a cutaneous neoplasm [35,36,37,38,39,40,41].

Over the past two decades, multiple studies have attempted to identify risk factors, host susceptibility, or UVR exposure factors [41,42,43,44,45,46,47] in UM development. Except for eye color, the relationship between most other factors has been inconsistent, if not contradictory. There is significant evidence that lighter (blue or gray) eye color is linked to a higher risk of UM. When compared to darker-colored eyes, lighter-colored eyes are correlated with a 75 percent higher risk of developing UM.

There are two hypotheses for this association. The first one claims that the eyes with a light-colored iris also have less melanin in the choroid and the retinal pigment epithelium [48], resulting in less UV protection. The second hypothesis indicates that a lighter-colored iris could be a predetermined phenotype for developing UM, regardless of melanin levels [9].

In contrast to findings that the “fair phenotype” is a risk factor for CM, investigations focusing on the link between host susceptibility characteristics and the UM have generated conflicting results. Light eye color, light skin color, and the ability to tan have been linked to the development of UM in several studies [9]. It is widely known that the incidence of UM is substantially lower in darkly pigmented people than in light-pigmented ones [49]. Even among white people, there is evidence that having a lighter skin color is linked to a higher risk of developing UM. When compared to people with a dark skin tone, fair-skinned people have an 80% greater chance of developing UM. It is considered that white people with lighter skin have less or different types of melanin protecting their eyes, increasing the risk of developing UM. Fair skin could potentially be a surrogate marker for a susceptible genotype, regardless of the quantity or type of melanin pigment.

There is a statistically significant connection between the capacity to tan and the risk of UM, with individuals who sunburn easily reporting a 64% higher chance of developing UM than those who tan well. This association is likely related to the previously postulated relationship between light skin color and UM.

Although various facts suggest the possibility of a fair-phenotype predisposition to UM, an independent relationship between light hair color and melanoma could not be established. Hair color does not seem to be an effective surrogate marker for follicular melanin patterns. Follicular melanocytes and epidermal melanocytes are different in several ways, notably the follicular melanocyte’s lack of sensitivity to UVR. Furthermore, hair melanin biochemical properties and melanin subtypes are not strongly linked to human hair color [9].

In addition, both UM and CM can occur de novo but can also arise from pre-existing melanocytic lesions such as nevi or primarily acquired melanocytosis [50,51,52,53,54]. Moreover, the risk of CM increases with the number and pathologic severity of cutaneous nevi [55]. The presence of many (≥100) or atypical nevi has been identified as a marker for the development of CM [56,57].

An increased number of nevi, either cutaneous or ocular, has been investigated as a potential risk factor for UM, as well. Studies reviewing the number of cutaneous nevi reported by patients themselves, observed a slight increase in the risk of UM as the number of nevi increased [35,58,59]. There is also a correlation between the number of clinically dysplastic cutaneous nevi and a higher risk of developing UM [39]. Particularly, patients with four or more atypical nevi have a substantially higher incidence of UM compared to those who had none [10]. Studies have confirmed that host susceptibility factors such as dysplastic nevus syndrome, ocular melanocytosis, and xeroderma pigmentosum may contribute to the development of UM, but a pre-existing iris or choroidal nevus is of particular concern [41,60].

The histopathologic classification of melanoma not only encompasses histologic characteristics but also contributes crucial information about the potential risk factors by providing detailed information about the cellular and tissue changes associated with these diseases. In the cases of both UM and CM, a histologic examination may reveal the development of the tumor on a pre-existing dysplastic nevus [61] and also underline the varying biological characteristics and behaviors exhibited by malignant melanocytic lesions depending on their unique genetic signature [62]. Other recent dermatopathology studies have helped described new entities, such as pigmented epithelioid melanocytoma, which are considered as having an intermediate malignant potential due to their infrequent ability to metastasize to distant sites, despite having a high potential to metastasize to regional lymph nodes [63].

### 3.4. Environmental Risk Factors

Although both CM and UM develop from melanocytes emerging from neural crest cells, the driver mutation environment for these two neoplasms is significantly different. This disparity can be explained by the absence of UVR in UM pathogenesis, whereas UVR is a widely recognized risk factor for CM [1].

CM can emerge anywhere on the body but is most common in areas chronically exposed to the sun [64]. The most significant environmental risk factor for CM oncogenesis is UVR from sunlight exposure [65,66,67], which induces melanocyte transformation by triggering DNA damage to the skin [33,68,69]. Exposure to UV-B spectrum radiation has been proven to consistently increase the risk of developing CM [70]. In addition, sun exposure timing and frequency have been linked to an elevated risk of CM in several studies [66,67,68].

For instance, compared to the chronic regular pattern of sun exposure, which is more typically associated with actinic keratosis and non-melanoma skin malignancies, the occasional, yet intensive sun exposure (characteristic of sunburn history) is associated with a higher risk of developing CM [68,69,71,72]. There is also evidence of increased occurrence rates in fair-skinned individuals with considerable UV exposure [66] and multiple sunburns, especially during the early stages of life [69].

The connection between UVR exposure and the development of OM, on the other hand, is less well-documented and often disputed. Still, due to the relationship between CM and UVR exposure, the disproportionate number of UM patients with blue eyes, and the rare occurrence of cutaneous and uveal melanoma in non-white patients, sunlight exposure has been evoked as a risk factor for UM [67,73]. UM is significantly more frequent in Caucasians, especially those with light-colored eyes frequently exposed to UVR, suggesting a correlation to UVR exposure [35,46,74,75].

An increased choroidal pigment seems to provide a UV-protective effect, thus explaining the extremely low incidence of UM in non-white races [73]. Furthermore, increased sunlight exposure of the lower half of the iris has been involved in the occurrence of iris melanomas [76]. Further investigations have rather linked short-wavelength blue-light exposure to the genesis of UM [77,78]. Blue light, a component of the visible light spectrum, penetrates more into the eye structures, causing significant damage to the retina.

Although only a portion of blue light is detrimental, it seems like we are becoming more and more exposed to it due to the widespread use of digital gadgets and modern lighting emitting blue light [1]. Moreover, similar to CM, the pheomelanin pigment pathway may contribute to UM genesis through an oxidative damage-dependent, UVR-independent, carcinogenic mechanism [79]. People with a pale complexion and light-colored irises are more likely to have CM and OM than people with dark-brown skin and dark-colored irises [77]. The evidence that, unlike CM, the prevalence of UM is not escalating during the last decades is also noteworthy [10].

### 3.5. Simultaneous Occurrences of CM and UM

Even though cutaneous and ocular melanomas are assumed to be inherited autonomously, the incidence of concomitant melanoma presentation as double primary tumors in the same patient and different members of the same family has questioned the possibility that these two types of melanomas are etiologically related [80]. Some studies suggested that individuals with atypical nevus syndromes are more predisposed to developing both cutaneous and ocular melanomas [10,39,81]. The atypical mole syndrome (AMS) [81,82], the dysplastic nevus syndrome (DNS) [83], the B-K mole syndrome [84], and the familial atypical mole and malignant melanoma syndrome (FAMMMS) [85] describe the presence of clinically atypical nevi with a family history of melanoma. A dominant genetic predisposition is suspected when UM occurs more than once in a single family, or when bilateral UM occurs in a single individual or with CM and multiple atypical nevi [86,87]. Other clinical features of family members with these syndromes include an increased risk of CM and multiple primary melanomas, an early diagnosis, and the occurrence of clinically atypical nevi (“dysplastic nevi”) exhibiting architectural ailments and differing extents of cytological atypia [88].

Atypical cutaneous nevi, mainly composed of melanocytes overstimulated by excessive exposure to UVR, have the potential to evolve into CM and are correlated with an increased risk of UM [10,39]. Patients with atypical moles are 10.4 times more susceptible than the general population to develop UM [53]. Considering that a correlation between primary OM and CM is suspected, there are no absolute guidelines that address whether patients diagnosed with OM should have dermatologic evaluations or, conversely, whether individuals diagnosed with CM should have ophthalmologic evaluations [9]. Patients diagnosed with OM must benefit from a cutaneous inspection performed by a dermatologist, especially if they have characteristics supporting AMS. Patients with more than four atypical nevi, a personal history of multiple sunburns at a young age, or a family history of CM should be educated about the clinical features of the disease. Once a year, dermatologic screening is indicated in patients with more than four atypical nevi, a personal history of multiple early sunburns, or a family history of CM. Routine ocular examinations are not considered cost-effective for all patients with atypical nevus syndromes [10], although the incidence of UM increased with the severity of the AMS phenotypic expression [39]. Routine ophthalmic evaluations are recommended for persons with a high expression of the AMS phenotype, iris nevi, and CM. Current data does not support the need for ophthalmologic evaluation in patients diagnosed with CM in the absence of a family history of CM or AMS [10], excepting high-risk families or if visual abnormalities are noticed [89].

### 3.6. Survival Rates

Survival rates in uveal and cutaneous melanoma differ based on several factors such as the stage of the cancer, the size and location of the tumor, and the patient’s overall health. However, overall survival rates are generally better for patients with cutaneous melanoma than for those with uveal melanoma.

According to the American Cancer Society, the 5-year survival rate for patients with UM is about 85%. In cases where melanoma remains confined to the ocular region, the 5-year relative survival rate is approximately 85%. Conversely, individuals afflicted with melanoma that has metastasized to neighboring tissues, organs, and/or regional lymph nodes exhibit a 5-year relative survival rate of 66%. In instances where the melanoma has extended to distant sites, the 5-year relative survival rate drops to 15% [90].

In contrast, the 5-year survival rate for patients with CM is about 94%. For individuals diagnosed with “thin melanoma” (less than 1 mm in maximal thickness) that has not spread to regional lymph nodes or other distant sites, the 5-year relative survival rate in the United States is estimated to be 99%. In the case of thicker melanomas, the 5-year relative survival rate may be 80% or higher, depending on several factors including the extent of tumor invasion and lymph node involvement. In the United States, the 5-year relative survival rates for melanoma that has spread to regional lymph nodes is 71%. When melanoma disseminates to distant organs, the 5-year relative survival rate declines sharply, typically to around 32% [90].

However, it is important to note that this figure is not universal and may vary considerably among individual patients, influenced by diverse clinical and pathological features such as the number of involved lymph nodes, genetic alterations, and the primary tumor characteristics such as thickness and ulceration [91].

According to epidemiological research, sex is an autonomous prognostic factor of CM even after adjusting for other established predictive markers such as age, Breslow thickness, ulceration, histologic subtype, location, and sentinel lymph node positivity.

The underlying reasons for these gender disparities have been attributed to behavioral and biological differences between males and females. Women have traditionally demonstrated better adherence to primary preventive measures such as UV protection and secondary preventive practices such as regular medical check-ups [92]. Although melanoma is not conventionally regarded as a hormone-responsive tumor, research suggests that androgen receptors are present on melanoma cells, possibly accounting for the more aggressive nature of this cancer in males. Moreover, estrogen may also have a role as data indicates that the survival advantage observed in females is attenuated in postmenopausal women with decreasing levels of estrogen [93].

In the context of UM, a notable gender disparity exists as well, as evidenced by the findings of some studies. Specifically, female patients exhibited superior overall survival during the first ten years after being diagnosed with UM but experienced a considerably higher incidence of UM-related mortality in the subsequent period. One possible explanation for the comparatively poorer survival rates among women during the latter period may be deduced from the characteristics of the patient cohort that remained in the study. Specifically, men experienced higher rates of mortality in the first decade, leaving behind a group of survivors who were relatively young, whereas women had lower mortality rates during this period, resulting in a group of relatively older survivors. Advanced age is a significant prognostic factor in UM, with a positive association between age and the risk of metastasis. This phenomenon may be attributed, at least in part, to a lead time bias, where older patients are more likely to have larger tumors that have been growing for an extended period [94].

## 4. Conclusions

Cutaneous and uveal melanoma are deadly forms of cancer arising from melanocytes. They are very distinct tumors biologically, exhibiting remarkable differences in terms of etiopathogenesis, however with some similarities between the high-risk population groups of these melanomas. Both malignancies derive from neural crest melanocytes which have migrated to the epidermic tissue or the eye. The morphologic and gross histopathologic examination of tumor material from both CM and UM demonstrates the common origin. Host susceptibility factors increase the probability of acquiring different subtypes of melanoma. Substantial, reliable, and consistent associations have been revealed between CM and the fair phenotype, with light eye color, light hair color, fair skin tone, and susceptibility to burns. Blue or gray eye color, fair skin, and the inability to tan are also identified as important risk factors in UM appearance. On the contrary, blonde or red hair has not proved to be a significant risk factor for the development of UM. While UVR is a significant risk factor for the development of CM and iris melanoma, it is not for the development of posterior UM.

There is consistent evidence for the association between UM, atypical nevi, and CM. This connection supports the recommendation that patients with a personal or family history of UM, particularly when combined with atypical nevi, should be evaluated for UM and CM on a routine basis.

In both types of melanomas, female patients appear to exhibit higher survival rates, and age is an influential factor that may act directly or indirectly on patient outcomes.

In recent years, tremendous information has been gained concerning the two types of melanomas. Although different in many aspects, UM and CM patient populations significantly overlap one another, therefore discussing the occurrence and characteristics of UM and assessing the available data on its potential association with CM could offer some guidance to dermatologists on the evaluation and diagnosis of this ocular malignancy.

## Data Availability

The data presented in this study are available on request from the corresponding author.

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
