# Peer review of "Differences and Similarities in Epidemiology and Risk Factors for Cutaneous and Uveal Melanoma"

_medicina, 2023, doi:10.3390/medicina59050943_

Round 1

Reviewer 1 Report

Please break up sections 1, 2, and 3 into smaller paragraphs. At present these sections are long run-on paragraphs that are basically unreadable. 

Author Response

Firstly, we would like to express our gratitude to the reviewers for their detailed feedback on our manuscript.

After carefully considering your comments, we have revised the manuscript and made all the necessary changes, as per your suggestions. I believe that the article has been significantly improved and is now ready for reconsideration. I have attached the revised manuscript, along with a point-by-point response to the reviewers' comments, for your perusal

We have implemented the following changes to the article: The sections „The cellular function of melanocytes”, "Epidemiology", " Host Susceptibility Factors", and "Environmental risk factors" have been broken down into smaller paragraphs for improved readability.

Additionally, we have restructured the article to better fit the format of a narrative review. A new paragraph has been added after the introduction to detail the review's methodology.

After careful consideration and discussion with all co-authors, we have incorporated histopathological examinations as suggested by some reviewers. We included a discussion on histology examination in the Chapter 3.3. as we found it relevant in identifying potential underlying risk factors.

The English grammar and spelling of the document have been checked. However, if further improvements are needed, we may consider using a paid editing service offered by the publication.

I hope you find that I have adequately addressed all the concerns and queries raised during the initial review process.

Thank you again for your invaluable input, which has helped to refine my work. If you require any further information or have any queries, please do not hesitate to contact me.

Looking forward to your favorable response.

Reviewer 2 Report

I thank the academic editor for giving me the opportunity to review this paper. It is a manuscript in which the authors conduct a literature review taking into account the differences and similarities between Uveal Melanoma (UM) and Cutaneous Melanoma (CM). I think the idea of looking at these two entities together, bringing out similarities and differences, is quite interesting. However, I feel, as an old pathologist, that much of the histopathology is missing, which the authors only mention in the conclusions.

Therefore, I would suggest to the authors to study, discuss and properly cite these papers concerning the histopathological part of melanoma and that would improve the quality of their manuscript:

1)Cazzato G, Arezzo F, Colagrande A, Cimmino A, Lettini T, Sablone S, Resta L, Ingravallo G. "Animal-Type Melanoma/Pigmented Epithelioid Melanocytoma: History and Features of a Controversial Entity. Dermatopathology (Basel). 2021 Jul 5;8(3):271-276. doi: 10.3390/dermatopathology8030033. PMID: 34287308; PMCID: PMC8293039.

2)Cazzato G. Histopathological Diagnosis of Malignant Melanoma at the Dawn of 2023: Knowledge Gained and New Challenges. Dermatopathology (Basel). 2023 Feb 13;10(1):91-92. doi: 10.3390/dermatopathology10010013. PMID: 36810571; PMCID: PMC9944108.

3)Alessio G, Guerriero S, Albano V, Piscitelli D, Falcone V, Lastella P, Resta N, Stella A. Neurofibromatosis type 1 and melanoma of the iris arising from a dysplastic nevus: A rare yet casual association? Eur J Ophthalmol. 2021 May;31(3):NP45-NP49. doi: 10.1177/1120672120906999. Epub 2020 Feb 16. PMID: 32064917.

Author Response

(The authors gave the same response as above.)

Reviewer 3 Report

In this paper, the authors conduct a literature review taking into account the differences and similarities between Uveal Melanoma (UM) and Cutaneous Melanoma (CM). The basic idea is not a bad one, as the authors set out, also quite extensively, to analyse various different factors underlying these two types of melanoma. I believe, however, that at least a major review is needed for the issues I will now list:

1) It is unclear what type of review setting the authors have conducted: is it a narrative review? is it a systematic review? is there a meta-analysis? It seems to me, more than anything else, that this paper can be considered a narrative review, in which the authors, starting from some data retrieved from the literature, discuss and argue the differences and similarities of these two forms of Melanoma. Please clarify this aspect and orient the structure of the manuscript in this sense.

2) It is not correct to state the materials and methods (how a review was conducted) in the introduction chapter. Therefore rearrange the structure of the paper.

3) In the discussion chapter on diagnostic aspects, please include indispensable histopathological information.

4) Please check the English and some typing errors.

Author Response

(The authors gave the same response as above.)

Round 2

Reviewer 2 Report

Paper can be accepted.

Reviewer 3 Report

Manuscript has been improved and now can be accepted